# Development and Demonstration of a Wireless Ultraviolet Sensing Network for Dose Monitoring and Operator Safety in Room Disinfection Applications

**DOI:** 10.3390/s23052493

**Published:** 2023-02-23

**Authors:** Michael F. Cullinan, Robert Scott, Joe Linogao, Hannah Bradwell, Leonie Cooper, Conor McGinn

**Affiliations:** 1Akara Robotics Ltd., D08 TCV4 Dublin, Ireland; 2Department of Mechanical and Manufacturing Engineering, Trinity College Dublin, D02 PN40 Dublin, Ireland; 3Faculty of Health, University of Plymouth, Plymouth PL4 8AA, UK

**Keywords:** disinfection robot, IOT sensor, healthcare, bluetooth low energy (BLE), ultraviolet (UV) disinfection, safety, wearable

## Abstract

The use of mobile ultraviolet-C (UV-C) disinfection devices for the decontamination of surfaces in hospitals and other settings has increased dramatically in recent years. The efficacy of these devices relies on the UV-C dose they deliver to surfaces. This dose is dependent on the room layout, the shadowing, the position of the UV-C source, lamp degradation, humidity and other factors, making it challenging to estimate. Furthermore, since UV-C exposure is regulated, personnel in the room must not be exposed to UV-C doses beyond occupational limits. We proposed a systematic method to monitor the UV-C dose administered to surfaces during a robotic disinfection procedure. This was achieved using a distributed network of wireless UV-C sensors that provide real-time measurements to a robotic platform and operator. These sensors were validated for their linearity and cosine response. To ensure operators could safely remain in the area, a wearable sensor was incorporated to monitor the UV-C exposure of an operator, and it provided an audible warning upon exposure and, if necessary, ceased the UV-C emission from the robot. Enhanced disinfection procedures could then be conducted as items in the room could be rearranged during the procedure to maximise the UV-C fluence delivered to otherwise inaccessible surfaces while allowing UVC disinfection to occur in parallel with traditional cleaning. The system was tested for the terminal disinfection of a hospital ward. During the procedure, the robot was manually positioned in the room by the operator repeatedly, who then used feedback from the sensors to ensure the desired UV-C dose was achieved while also conducting other cleaning tasks. An analysis verified the practicality of this disinfection methodology while highlighting factors which could affect its adoption.

## 1. Introduction

Ultraviolet disinfection systems have been used to reduce the bio-burden in a wide variety of applications, including water disinfection, upper air disinfection, ventilation systems, appliance decontamination, etc. [1]. UV-C radiation (wavelength: 200–280 nm) is typically employed. This is absorbed by the proteins, the DNA and the RNA in cells, leading to the break down of these structures [2]. The inactivation levels of micro-organisms achieved is typically expressed in terms of log reduction, where a 1 log reduction corresponds to an inactivation of 90%; 2 log, to an inactivation of 99%; and so on. The level to which micro-organisms are susceptible to damage from UV-C is dependent on their type. The required UV-C fluence to inactivate a micro-organism (referred to as a dose) is also dependent on the medium in which the micro-organisims exists (air, surface or water), as these provide different levels of protection [1]. The UV-C dose is typically expressed as a fluence (J/m^2^), which is the time integral of the UV flux (W/m^2^) from the UV-C source. Several tabulated lists of the UV-C dose required to reach desired inactivation levels are available [1,3].

Mobile platforms with UV-C lamps that can be positioned in rooms for the decontamination of the surrounding surfaces have become increasingly common, particularly in healthcare settings where their need is most acute [4]. These are found in several forms, from simple push-in-place devices to autonomous mobile robots (AMRs) that move through the environment automatically, emitting UV-C. The efficacy of such devices depend on a number of factors that effect the UV-C dose delivered to surfaces [1,5,6]. This is not only dependent on the lamps rated power but also on the age of the lamps, the presence of any dust or other residue, and, in the case of low-pressure mercury lamps, the length of time the lamps have been turned on, as they do not emit their steady-state output until they reach operating temperature. The position of the UV-C source within a room affects the fluence delivered to surfaces as the flux decreases according to the distance from the lamp, following an inverse square law. Additionally, shadowing will mean that in any one location, the optical radiation can be blocked from some surfaces in the room. This aspect is particularly problematic for UV-C, as the reflectivity of most surfaces is very poor in this part of the spectrum, meaning reflection cannot be relied upon to deliver optical irradiation to surfaces [7]. Naturally, the length of time the robot spends in each location affects the total fluence delivered. The incident angle between the optical irradiation and the surface is also relevant, as any deviation from the perpendicular increases the effective area over which the optical irradiation is incident. Other environmental factors, such as humidity, can also have an effect [1]. Practical considerations also play a role, as the device must be convenient to use and integrated into a healthcare facilities operational procedures so as to make it practical for staff to benefit from its use.

To ensure the effectiveness of UV-C disinfection systems, it is necessary to estimate the UV-C dose delivered to the surfaces, air or water to be treated to ensure the dose is sufficient to achieve the desired inactivation levels for the target micro-organisims. In typical applications (such as water treatment plants and ventilation systems), this has been achieved by calculating the dose delivered based on the lamp power and distance from the target [8]. However, in a case where the UV-C device is mobile and operates in unstructured environments, a more reactive method is required. Some devices have used an operating procedure that positions the UV-C emitting device in the room for a time that was calculated to deliver the required UV dose with some margin for error [9]. This approach was time and energy inefficient and did not provide any validation that the required dose had actually been achieved. It also did not consider changes in the room, such as additional furniture, etc. Other approaches have used a measurement of reflected UV-C to estimate the flux reaching the surfaces (for example, the Tru-D platform) [10]. The amount of reflected UV-C detected by the robot was a function of the room size and the types of objects in the room. However, surfaces that reflect UV-C more effectively, such as polished metals, distorted the reading. This method also required the use of very powerful lamps, so there was sufficient reflected UV-C to be detected. Measuring the geometry of the room was also used to estimate the UV-C dose at surfaces based on the lamp power and distance to those surfaces (for example with the Surfacide platform) [11]. This did not, however, consider lamp degradation, environmental factors, etc.

A number of sensing technologies were employed to measure UV-C levels for validation and safety [12]. Spectro-radiometers measure irradiance at multiple individual wavelengths, allowing for the analysis of sources to determine the spectral weighting of the optical radiation they emit. While this type of sensing can be very accurate and provides rich information about the source, they are generally too large and costly for use in the field. Hand-held detectors can be divided into broadband and narrowband types. Broadband detectors allow for individually calibrated filter combinations to be used to weight the relative response to different wavelengths in a range, for example, to mirror the weightings used for occupational safety limits. Narrowband detectors (for example, the Omega HHUV254SD [13]) are most commonly used in UV-C disinfection applications as UV sources, such as mercury lamps and LEDs, emit the vast majority of their radiation over a small range of wavelengths. These devices are most commonly hand held, with a probe and readout device, but there have also been a number of detectors designed to be positioned in a room and read wirelessly or periodically. Devices, such as that from GenUV [14], UVCense [15] and Promax [16], relied on Bluetooth low energy (BLE) to report the recorded UV-C flux and fluence. Others, such as that developed by L&M instruments [17], log the data locally, which could then be retrieved after the disinfection operation. An alternative to electronic sensing was the colourimetric indicator paper that changed colour in response to UV-C irradiation. These have been used in a number of studies to verify the dose delivered to surfaces during a disinfection procedure [18,19,20]. This approach relied on the comparison of the colour on the paper with a reference palette and, thus, had a high degree of uncertainty.

The safety of operators is of paramount importance in the use of UV-C technologies and is the reason why most room decontamination devices require the operator to leave the room during their use. UV-C irradiation is known to be harmful to mammalian skin and eyes, but the effect is dependent on the wavelength, with shorter wavelengths having more energy, but being more readily absorbed and so only penetrating the upper layer of the epidermis [21]. Here UV-C can cause erythema, with red inflamed patches on the skin [22]. The main chronic effect of UV-C in the eyes is photokeratitis (also known as snow blindness or welder’s flash) which results in severe pain and an inability to use the eyes for one to three days [23]. It has been noted that the number of studies relating to the long-term effects of UV-C exposure is insufficient to draw strong conclusions [24]. The US FDA advise that most skin irritation resolves within a few days and the risk of skin cancer is very low [25]. A number of standards exist for the occupational safety limits of workers who are exposed to UV-C. EU directive 2006/25/EC regulates the safety of optical radiation. The limit is weighted based on the wavelength of the radiation, with an effective radiant exposure (H_eff_) being used to express the wavelength-weighted sum of the radiated flux a person is exposed to in the 180–400 nm range. The weighted daily exposure limit is 30 J/m^2^. Typical low-pressure UV-C lamps emit optical irradiation at 254 nm, the weighting of which is 0.5, meaning if a worker is exposed only to UV irradiation from this source the unweighted daily limit is 60 J/m^2^. This is the same value as appears in ISO15858 [26].

While UV-C decontamination robots have enjoyed success in several healthcare settings, there exists a need to better integrate with the workflow of cleaners to increase the applicability of this technology to additional environments. Developing procedures and systems to allow UV-C disinfection to be conducted at the same time as other cleaning and room preparation procedures has the potential to reduce the overall time between the clinical use of rooms, increasing overall hospital efficiency. In addition, such procedures must be validated to ensure a sufficient UV-C dose was delivered to surfaces in the room during each procedure to maintain quality. Additionally, it must be practical for robot operators to quickly introduce robotic procedures to rooms, minimising arduous setup procedures. This work builds upon the sensor design described in [27], and presents a wireless sensor network with the aim of addressing these failings of current UV-C technology. These sensors aim to monitor the UV-C flux incident on a surface for the purpose of feedback and validation in real-time, allowing disinfection procedures to adapt to real-world conditions. The same sensors are also used, in a different configuration, to monitor the exposure of an operator to UV-C, protecting them from exceeding the occupational safety limits. To evaluate the suitability of these sensors to support validated disinfection procedures the sensors are integrated with a robotic disinfection platform. The procedure is trialed in a clinical setting, with both sensor use cases being utilised, delivering a validated UV-C dose while allowing an operator to remain in the room safely.

## 2. Materials and Methods

The requirements for the UV-C sensors were first established based on the outcomes of previous research [27], the desired operating procedure as developed with clinicians, experts and technical analysis of the problem. The summarised design objectives of the sensors were:The sensors should be able to detect and measure UV-C, in particular around the 254 nm range which is most commonly used for room disinfection.The sensors should be easily secured to a diverse range of surfaces, and allow for multiple orientations.The sensors must be convenient to use by an operator.Radiation over a large incident angle must be accurately recorded.The sensors should be capable of being monitored by common devices such as smartphones, laptops etc as well as specialised UV-C disinfection hardware.Multiple sensors should be monitored simultaneously and in real-time to inform the disinfection procedure.The sensors should be sufficiently low-cost that it is practical to use several sensors on a regular basis.If the connection between the sensor and the device reading from it is lost, this should not affect the accuracy of the UV dose measured.

The same sensor design should be capable of monitoring the UV-C exposure of an operator for safety purposes. This has the additional requirements of:The sensor must be able to be attached to the user’s body.The sensor must signal the user directly when UV-C is detected, and not rely on the response of a connected device.After being signaled, the user should have the means to confirm that they have moved to safety and reset the device.The sensor should be sensitive to small levels of radiation, down to 0.002 mW/cm^2^.Platform integration should allow the user to see how much UV-C they were exposed to over a daily period. It should also stop UV-C being emitted from the robot if the user is exposed to a threshold UV-C flux or does not take action to remove themselves from the area of UV-C exposure.

While some of the UV-C radiometers mentioned in the previous section could fulfil the requirements of the monitoring use case, the operator safety monitoring function requires bespoke hardware. Furthermore, these devices are too expensive and difficult to use for routine disinfection procedures [19]. They have not been designed for use as part of a control system, while colorimetric indicators have been used for measurement of UV fluence, this technique cannot provide real-time feedback.

### 2.1. Sensor Design

Bluetooth low energy (BLE) was chosen as a communication interface for the sensors as it allows information to be transmitted to multiple devices such as smartphones, laptops, etc. concurrently, without the need for specialist hardware. BLE requires relatively little energy to operate, with a single coin cell battery providing enough energy for up to 14.1 years of operation for simple devices [28]. BLE also has the capability to operate in mesh networks, and can be used for localisation, both of which are advantageous for further development of this work. A readily available BLE module (model: ESP32 WROOM [29]) was chosen due to its low cost and community support. This module provides the logic elements of the sensor in addition to BLE functionality.

A UV-C sensitive photodiode (model: GUVC-S10GD [30]) is used as the sensing device. This produces a small electrical current in response to UV-C radiation. The GUVC-S10GD gives a viewing angle of 150 degrees which is advantageous for detecting UV-C irradiation with a large incident angle. The signal is amplified and converted to a voltage using a transimpedance amplifier circuit. The output of this amplifier is read by a 16 bit analogue to digital converter (ADC) with a range of ± 2.048 V [31]. The built in ADC on the ESP32 was not used for this purpose as it has poor linearity and does not detect voltages below 70 mV. With the selected resistors used in the transimpediance amplifier (giving a gain of 8 million), this gives a UV-C range of up to 3.3 mW/cm^2^.

In addition to these core features, the sensors also feature a USB-C port, charging and protection circuitry for a 1000 mAh battery, as well as an RGB LED to indicate the state of the sensor, while the same sensor design is used for both the UV-C monitoring and safety sensor, the safety sensor makes use of a piezoelectric buzzer and an additional input button.

The mechanical design of the UV-C monitoring sensors is shown in Figure 1. The enclosure is designed to protect the sensor from mechanical damage and facilitate mounting while remaining as small as possible to reduce the surface area covered by the sensor which blocks the UV-C irradiation. The photodiode is exposed but is recessed behind the surface of the housing. The sides of the enclosure are orthogonal, providing several orientations in which the sensor can be placed on a flat surface. There is also a removable, rotatable hook which can be used to attach the sensor to handles, the edge of surfaces etc. This hook can also be pivoted to support the sensor at an arbitrary angle on a flat surface or to support the sensor on uneven surfaces such as a bed. The design of the safety sensor is similar but the input button is available in the centre of the sensor. This is secured to a chest mount allowing the user to comfortably wear the sensor for long periods with easy access to the button. This design, with a single front facing sensor will be incapable of detecting UV-C irradiation outside the field of view of the sensor. However, the greatest risk from the UV-C source is to the face and eyes of the wearer and this approach avoids the additional complexity of a multiple sensor configuration.

The sensors printed circuit boards (PCBs) were manufactured and assembled with the majority of the components at a cost of less than €12 each (quantity 20). The photodiode and battery cost an additional €7 and €8, respectively, for an overall cost of €27 before the enclosure is accounted for.

### 2.2. Functional Operation

While the monitoring sensor is in operation, readings are requested from the ADC four times per second. It is not expected that the UV-C flux will change quickly and this simple approach simplifies the sensors operation. This value is averaged and converted to a value of UV-C flux incident on the sensor. As the experiments below show, the linearity of the sensor is good in the area of interest and so a linear calibration curve is used. The change in time since the last measurement was taken is then used with the flux value to update the fluence estimate. Calculating and reporting the fluence rather than just the flux negates the possibility that an intermittent connection to the connected device decreases dose accuracy and allows sensors to only be monitored at the end of the UV-C use if so desired. An elapsed time value is also updated, which is the time since the sensor was last reset (when the fluence was set to 0 mJ/cm^2^). The sensor is reset at start up but it can also be reset using the BLE GATT (Generic Attribute Protocol) as explained below. Additionally, the ESP32’s onboard ADC is used to measure the voltage on the LiPo battery (via a voltage divider) and this is used to estimate the battery charge. The BLE interface is updated with this data.

The safety sensor operates in a similar way, but with the addition of a buzzer, which sounds when the fluence recorded exceeds 0 mJ/cm^2^. This continues to sound until the operator moves away from the UV-C source and presses the button located on the front of the sensor which resets the fluence to 0 mJ/cm^2^. This requirement for the operator to acknowledge the UV-C exposure is intended to ensure that even short exposures are noticed.

#### BLE Interface

The BLE interface is used to transmit data from the sensor. It can also be used to reset the sensor. The GAP (Generic Access Protocol) is used to advertise the presence of BLE devices. For BLE4.0, the advertisement package contains up to 31 bytes with an optional additional scan response package of the same size which is sent on request. The package contains information about the device identified with flags such as the devices unique address, transmitted power level etc. The manufacturer specific data flag is used in this case to broadcast the flux, fluence, time elapsed since last reset, and the battery level. The structure of this portion of the GAP profile is shown in Figure 2. The use of the GAP protocol in this way allows multiple devices to read the status of all of the sensors in range.

The GATT interface can also be used to read the UV-C flux and fluence values from the sensor. All of the values represented in the GAP, with the exception of battery level can be read from the sensor as characteristics once a connection to a central device is established. In addition a write only characteristic is provided to reset the sensor (return the fluence value to 0 mJ/cm^2^).

### 2.3. Sensor Validation

A number of tests were conducted to understand the operating conditions and limitations of the sensors. For these tests, a radiometer (model: Omega HHUV254SD) was used as a reference against which to evaluate the sensors. The UV-C optical radiation source used is a Philips TUV PL-L 55W/4P HF low pressure mercury lamp, while the output power of the lamp is not adjustable, the UV-C irradiation incident on the sensor can be modified by moving the lamp relative to the sensor. The lamp, sensor and reference radiometer were turned on for 20 min prior to the start of the experiment to ensure they had reached normal operating temperatures. All experiments were conducted at room temperature.

#### 2.3.1. Linearity

An ideal sensor will have an output that varies ratio-metrically with the quantity to be measured. A sensor that behaves like this is more easily calibrated as only two calibration points are needed. The sensor was positioned with the HHUV254SD probe as close as possible to the photodiode and directly facing the lamp. The lamp was moved through a distance of approximately 4.5 m to 1 m from the sensor to vary the UV intensity. Figure 3 shows the apparatus involved in the procedure.

#### 2.3.2. Cosine Response

According to Lambert’s Cosine law the irradiance intensity at a surface should vary with the cosine of the incidence angle. A sensor designed to monitor the irradiance at a surface would ideally vary in this way. To test this, the sensor and the reference instrument were positioned approximately 1.5 m from the optical radiation source directly facing it, which resulted in a radiance flux of 0.34 mW/cm^2^. From here the sensor was rotated each direction in increments of 10° up to 90° from the initial position directly facing the source. The reference instrument did not move and was monitored to ensure the radiance flux at 0° did not change. The experiment was repeated with the sensor in a horizontal and vertical orientation.

### 2.4. Robot Integration

The robot disinfection platform used in this research was developed by Akara Robotics [32] and is shown in Figure 4. This robot is designed for disinfection in hospital settings. It uses three 75 W low-pressure lamps (model: Philips TUV75WHO1SL) and can direct the optical radiation produced towards areas of interest in its vicinity, by rotating the column which houses the lamps and a reflector. While low-pressure lamps are used, alternative UV-C sources are equally as applicable to this methodology, although the sensors may require re-calibration if a different wavelength is produced. In contrast to most room disinfection devices, the design of this robot is intended to allow the operator to remain in the room while the disinfection procedure is being conducted. This is achieved by directing UV-C away from the operator and monitoring the location of the operator and occupancy of the room using a suite of cameras with a person detection system. Allowing the operator to remain in the room as the robot operates allows manual cleaning or room preparation to be conducted at the same time, reducing the downtime of the room. The robot makes use of the robotic operating system(ROS) which makes integration of new features, such as the sensors developed here, more feasible.

A number of operating modalities are envisaged for the use of the sensors developed in conjunction with this robot. For autonomous disinfection applications, where the robot automatically navigates through its environment irradiating selected surfaces, the monitoring sensors can be used to validate the UV-C dose delivered to points in the room. The ability to send real-time UV-C measurements to the robot allow it to respond to small changes in the environment, such as errors in the robot position, changes in furniture placement, etc., to ensure a consistent dose is provided.

Manual operation of the UV-C device, where the robot is moved into place by an operator and the UV-C lamps are manually controlled by them, is also an operating mode of this device with some important advantages. In terms of risk mitigation, there is little possibility for the robot to move and expose the operator to unsafe levels of UV-C. Some environments may be cluttered or frequently reconfigured, making autonomous navigation impractical. It also facilitates collaboration of the operator with the robot, repositioning furniture etc, so that the UV-C can reach additional surfaces. In addition, the setup for each individual room for autonomous operation may be arduous, given the number of rooms and the requirement to map them prior to operation.

There are two main manual operating procedures that can benefit from the use of the sensors. Firstly, a room can be analysed prior to the deployment of the robot, and a strategy developed for the way in which the robot should be moved through the environment, the surfaces treated, the target dose etc,. In this case, the position of each sensor would also be planned, allowing this to be paired with the sensor readings, giving a breakdown of the fluence at different points in the room. Alternately, the use of the sensors can assist in ad hoc disinfection of rooms, where little analysis is done on the room prior to use of the robot. Here an experienced operator can place sensors at points perceived as being strategically important within the room, and the feedback from the sensors used to inform the operator’s decisions regarding when to adjust the robot. This approach allows the robot to be quickly deployed in many rooms, and so can be beneficial in settings such as hospital wards. In both cases the UV-C fluence data from the sensors, not only informs operator decisions, but is also logged so it is available for review by management for auditing purposes.

A Silicon Labs BLED-V1 BLE dongle is used to provide the BLE interface on the robot. A ROS node monitors this and reads data from the sensors when they are available over GAP. The unique ID associated with each BLE device is used to identify each individual sensor and match this with a name from a database and also the type of sensor (safety or verification). The information from all the available sensors is published on a ROS topic so it can be accessed by other nodes on the system. This topic also records the last time the sensor advertisement was received from each sensor, so subscribers to the topic can know if the connection has been lost. Two methods to reset the sensors are provided. A ROS action connects to each selected sensor and resets them using the GATT. This can however be quite time-consuming when a large number of sensors are used. A ROS service is also provided to reset the sensors on the robot only (i.e., by applying a correction factor which returns the fluence to 0 mJ/cm^2^ at that time). The safety sensor cannot be reset in this way as it is desired that this type of sensor can only be reset by the operator at the sensor. To the contrary, when the safety sensor is reset by the operator the previous fluence value is retained and applied to data published by the ROS node so that the total exposure over the operating period is available.

A user interface on a 10.1″ touch screen, located at the rear of the robot, was adapted to display the data from the sensors. This is shown in Figure 5. When the robot is being configured for a decontamination procedure, a target UV-C dose can be selected using a slider. To inform this decision, a list of UV-C doses corresponding to log reductions of frequently targeted micro-organisms is provided. For rooms where a detailed formal standard operating procedure has been developed, a pre-configured UV-C dose is loaded. In this case, a list of locations where the operator should place each sensor is displayed, which increases the utility of the recorded data. When UV-C flux is received at a sensor a progress bar starts filling corresponding to the percentage of the target dose received at that sensor. A time estimate is provided which predicts the time remaining for the target dose to be received, at which point the progress bar changes to green. The battery level of the sensors is also displayed. The safety sensor is displayed in yellow, and the progress bar displays the proportion of the occupational safety limit received. A “Start cleaning session” button resets all the sensors (using the correction factor approach) and starts logging functionality which records the total dose recorded by each sensor when the disinfection session ends.

Regardless of the robot operating modality the safety sensor functions in the same way. When low levels of UV-C are detected this is displayed on the user interface but no action is taken. This will of course trigger the audible warning on the sensor iteslf, giving the operator the ability to address the issue without disrupting operation. When the UV-C level exceeds a threshold (set to a flux of 0.01 mW/cm^2^ or a fluence of 0.4 mJ/cm^2^) the robot enters an emergency stop state and the lamps must be turned on again by the operator. Additionally if communication from the sensor is lost during operation (defined as more than 5 s since an advertising message is received), the robot will initiate an emergency stop.

### 2.5. Testing in a Healthcare Setting

The robot was deployed in a major UK hospital setting to evaluate the ability of the UV-C sensors to be incorporated into real-world disinfection procedures. The purpose of this trial was not to establish the accuracy of the sensing (as this has previously been tested), but rather the usability and practicality of such a system.

#### Manual UV-C Protocol

The robot was used as a secondary disinfection measure (used after the room was manually cleaned using a chlorine-based biocide,) during terminal disinfection for rooms, in the isolation ward of the hospital. Given the exploratory nature of this work and the limited time available to conduct the trial, the robot was operated by a researcher familiar with the device and the dangers associated with UV-C rather than a hospital cleaner or other member of staff. Prior to the robot’s operation, a risk assessment was conducted for the robot as well as an assessment of the rooms to identify any additional hazards present. A warning sign was placed on the only entry door to the room, to prevent anyone from entering the room while the robot was in operation. Personal protection equipment was worn by the operator, including an EN170 rated visor and nitrile gloves for protection from the UV-C.

For the purpose of this study, a manual disinfection procedure (where the robot is manually positioned by the operator) was employed as this operating modality has a greater potential benefit from the sensors due to the inherent reduced repeatability of a manual process. The ad hoc approach outlined above was used as the rooms to be disinfected could not be predicted ahead of time due to hospital scheduling. Prior to the trial, the operator consulted the infection prevention and control team in the hospital to inform the choice of sensor locations, although the exact location was at the discretion of the operator during the procedure. Seven monitoring UV-C sensors were used in conjunction with a safety sensor worn on the chest of the operator. For all trials a UV-C dose of 40 mJ/cm^2^ was targeted. During the disinfection procedure, the operator wore a camera so that the process could be subsequently reviewed. At the commencement of the procedure, sensors were placed at points of interest in the room and turned on. Communication with the robot was verified on the touch screen. A disinfection session was started using the user interface to start the logging process. The lamps were turned on, and the first disinfection point treated, while this was in process, the operator could conduct other tasks such as room cleaning and preparation until the desired dose was achieved. The robot was then adjusted (e.g., by rotating the lamps) or moved to another location and the procedure repeated. If the robot was to be moved significantly, the lamps were first turned off. Furniture in the room was also moved and/or rotated, to treat different parts of it or for robot access.

## 3. Results

### 3.1. Linearity

Figure 6 shows the results from the linearity tests from one sensor. The output units here is the raw output from the ADC. The fitted linear model shows good agreement with the measured values with a root mean square error of less than 0.001 mW/cm^2^. The error is further illustrated in Figure 6b which shows the error between the linear model and measured data for each measurement.

### 3.2. Cosine Response

The response of the sensor as it is rotated away from the UV-C source is graphed in Figure 7. The relative angular response is given by:(1)$$r_{\theta }=\frac{\phi }{\phi_{0}cos\left(\theta \right)},$$
where θ is the angle at which the sensor has been rotated from directly facing the UV-C source at θ=0, ϕ is the UV-C flux measured by the sensor and $\phi_{0}$ is the measured flux when θ=0.

### 3.3. Results from Hospital Trial

Six disinfection procedures were conducted, four of which were recorded with a chest-mounted camera on the operator. An example of the room layout and the procedure followed is shown in Figure 8. Additionally, the videos were analysed to provide a breakdown of the activities conducted during the disinfection procedure (Figure 9). Information on the UV-C exposures detected by the safety sensor, including as a percentage of the occupational daily safety limit (6 mJ/cm^2^) is shown in Table 1, along with the total time taken for each procedure.

## 4. Discussion

The sensors developed successfully demonstrated the capability of a sensor-based approach for enhanced UV-C room disinfection. The results of the linearity tests show strong agreement with a linear model over the measured range. The residual errors observed are within the margin of error for the experimental procedure. The accuracy of this model justifies the use of a two-point linear calibration procedure for the sensors which converts the raw ADC reading to UV-C flux.

The cosine response test results show a favourable response up to approximately 60° in each direction of rotation, at which point the measurements reduce more than predicted with the cosine model. It should be noted that at larger angles, the UV-C flux value becomes small, and so this is less significant for the purpose of UV-C dose monitoring. However, for the UV-C safety sensor it is desirable that UV-C irradiation incident from any angle would be detected. If this is compared to commercial sensors as tested in [12], the sensor performs better than other sensors which do not use diffuse optics to increase their angular range. The use of such optics would be a useful improvement for the safety sensor.

Despite the relatively short deployment of the UV-C robotic device a number of important findings were made. The sensor system integrated with the robot well and there were no communication issues apart from some user error turning the sensors on. The operator was able to place the sensors appropriately within the room, making use of the flat sides and the hook, which was especially useful on the sides of beds and other pieces of equipment. The use of the sensor interface did not present any significant challenges for the operator, and it was convenient for them to identify when the required dose had been delivered. The procedure followed changed somewhat during the trial as the operator gathered experience and was able to position the sensors and robot with greater efficiency. In addition, in “Disinfection 1” there was no attempt to reposition furniture in the room for additional disinfection, but this was added subsequently. This procedure allowed both sides of mattresses, tables, cabinets etc. to be disinfected, a possibility not presented by most room UV-C disinfection protocols. The value of the sensors is also evident when examining the number of times the lamps are turned on and off during the procedure. As low-pressure lamps require some time to reach their optimum operating temperature, the UV-C output is difficult to predict. The use of sensors in the environment removes this as a factor.

The time measurements taken from the recordings of the procedures should be viewed as indicative of the distribution of tasks for a room not treated by the operator previously. The decision on where to place sensors and position the robot was made by the operator during the procedure which, in some cases, took considerable time, and more frequent adjustment than would be anticipated in subsequent procedures. A more structured approach to this, whereby the sensor positions are predetermined, would increase efficiency and improve the consistency of disinfection procedures, especially when different operators are considered. Seven monitoring UV-C sensors were used during this trial. From the procedure used by the operator this was insufficient, which resulted in the operator moving sensors between locations. In these cases, the sensors were not reset, and so the operator would calculate what sensor output would be necessary to reach the target dose (i.e., by adding the required dose to the initial value). This is an unacceptable cognitive load and should be addressed by the addition of more sensors and/or the ability to easily reset these sensors during the procedure. Additionally, this technique meant it was impossible to audit the logs for the UV-C doses delivered as multiple doses were combined. It is also interesting that there is significant time available during the disinfection process when the operator’s time is available for other activities such as traditional cleaning practices or room preparation. It is reasonable to assume that this available time would increase with the experience of the operator, allowing more room preparation work to be conducted during the UV-C procedure, reducing the overall time for room turn around. Care must be taken however in the placement of the robot and the desired area of the room for the operator to be active. Several times during the disinfection procedures the operator was unable to access much of the room because the robot’s lamps blocked access.

The need for additional guidance and the development of best practices was evident from the tests, in particular, sensor placement should be refined. It is more accurate that the sensor be placed parallel with surfaces but in some situations, particularly on horizontal surfaces, the sensors were placed on their edge, to face the UV-C source. This would result in a larger measured fluence. Similarly, the sensor should be placed on the part of the surface furthest from the optical radiation source to ensure that all of the surfaces received the target dose. In addition, it should be noted that the part of the surface covered by the sensor will not be exposed to UV-C and so the sensors should be placed close to, but not directly over, the most critical areas to treat, at a point with the same radiation exposure characteristics. It was also observed that there seemed to be an occasional tendency of the operator to move the robot closer to the sensors in order to decrease the time required to reach the target dose. This reduces the area over which UV-C is applied, as some surfaces will no longer be in the irradiation field of the lamps, and so will not be treated. This change of priority from ensuring appropriate disinfection of surfaces, to making the sensor record the required dose can be addressed with improved operator guidance.

Some issues were identified with the design of the sensor. The robustness of the hook mechanism was insufficient and resulted in some failures during the trial and in subsequent use. A related issue with the hook was its inability to fit around certain mounting points such as large bars on bed structures. This resulted in increased difficulty for the operator and a longer time to place the sensor. Additionally, the power switch on the sensors failed on some of the devices. This is likely due to the use of gloves which makes it more difficult to perform the fine motions needed to turn the sensor on/off.

The safety sensor successfully identified operator UV-C exposure. The majority of these exposures were at low flux levels, generally as a result of reflections in the environment. In particular, when the robot operates in the bathroom (a confined space) the frequency of exposure was greater. When the sensor received UV-C flux and started emitting the audio warning, the operator was able to respond and move out of the way within seconds. In some cases, generally when the operator moved to the side of the robot, a larger UV-C dose was experienced and the UV-C lamps were turned off by the robot. This form of user error could be minimised with greater experience. In this way the safety sensor can act as an educational tool, alerting the operator to aspects of the procedure they follow which are likely to result in UV-C exposure so they can be avoided in future. In general, the recorded level of UV-C exposure was low, and even for the highest UV-C dose recorded for a disinfection session (“Disinfection 6”, during which the operator was exposed to 5.23% of the limit over a 39-min period) this would require over 8 h of continuous robot operation to reach even 65% of the UV-C exposure limit, while the safety sensor has been shown to provide useful feedback to an operator, and steps have been taken to reduce the likelihood of an unsafe failure, the operating principles of the device do not meet the requirements of international standards for safety-related parts of a control system, e.g., ISO13849 [33]. Further development and validation of the technology is required to meet this standard.

## 5. Conclusions

The ability of environmental UV-C sensing to contribute to the operation of a room UV-C disinfection device was demonstrated here. A low-cost sensor was developed which showed good performance characteristics in terms of accuracy and cosine response. When integrated with a UV-C disinfection device, these allowed an operator to ensure they had delivered a sufficient UV-C dose to the surfaces in a room, while also providing a means to audit the procedure. Additionally, the sensors were used to enhance the disinfection procedure by allowing an operator to safely remain in the room with the UVC source and manipulate the furniture for better UV-C coverage. This also carries the benefit of enabling UV-C disinfection to be conducted alongside traditional cleaning, reducing the overall time required compared to both procedures being conducted sequentially. This does however rely on determining correct protocols to reduce the incidents of any UV-C exposures to minimise operator disruption. As the risks associated with regular doses of UV-C below the daily limit is uncertain, the combination of personal protective equipment and active sensing offers redundancy for applications such as this which seek to use specialised hardware with trained personnel to allow UV-C disinfection alongside a person.

The methodology described here enables faster, more robust decontamination of rooms and has the potential to increase the utility of UV-C disinfection in various healthcare settings. It may be particularly useful in high-turnover settings such as radiology. The insights gained will be incorporated into future work involving longer deployments of the technology and utilising hospital staff as the main operators of the system. While this work focuses on improving UV-C disinfection procedures using sensing technologies, several additional factors affect the adoption of UV-C robots in healthcare settings. These include the design of the built environment, the ease of integration of these devices with existing workflows, the importance placed upon environmental hygiene (including associated budget) and data-driven processes being greater utilised in this sector.

## Figures and Tables

**Figure 1 sensors-23-02493-f001:**
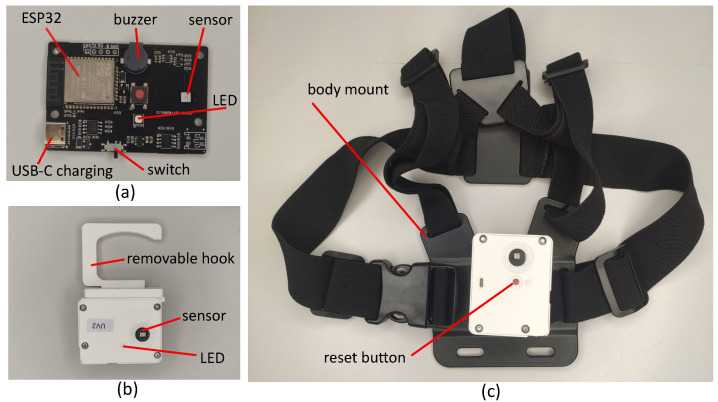
The sensors developed for this research. (**a**) The PCB used for the sensors showing some of the important features. (**b**) The sensor in the enclosure used for the monitoring sensors. (**c**) The safety sensor in the mount to be worn on the chest of the operator.

**Figure 2 sensors-23-02493-f002:**
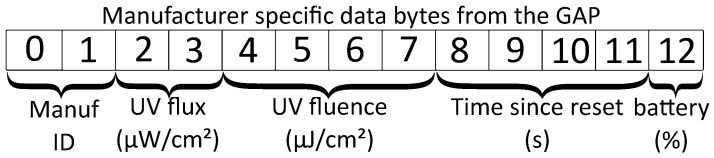
The manufacturer specific data sent as part of the GAP profile. All data with the exception of the manufacturer ID are integer values. The integers are represented in little-endian byte order.

**Figure 3 sensors-23-02493-f003:**
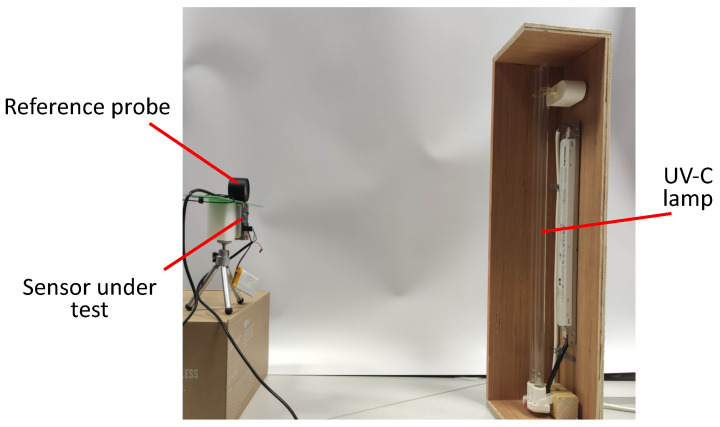
The testing apparatus for linearity testing and for calibration of individual sensors. Note that for clarity, the UV-C source is shown closer to the sensors than the minimum testing distance.

**Figure 4 sensors-23-02493-f004:**
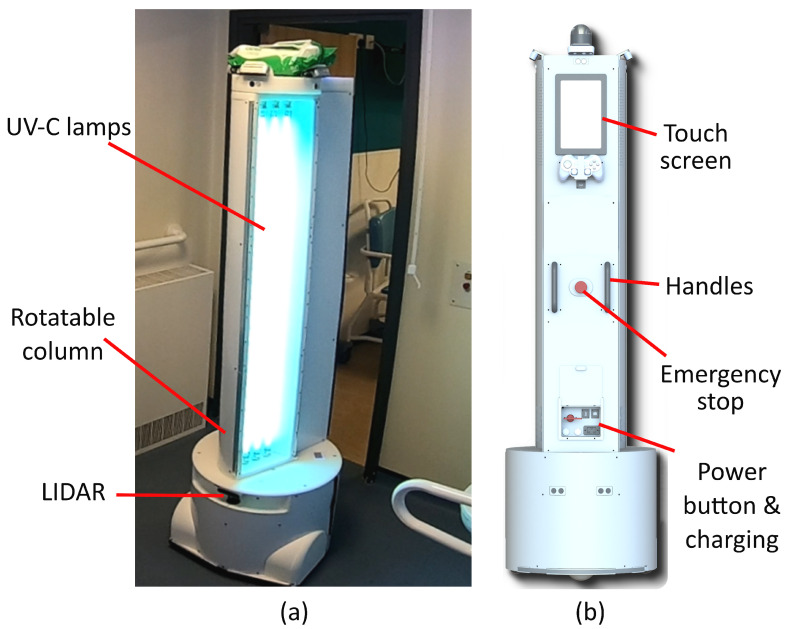
The UV-C disinfection robot used in the trial. (**a**) shows the robot in a hospital setting (**b**) shows a rendering of the rear of the robot where the touch screen interface is located.

**Figure 5 sensors-23-02493-f005:**
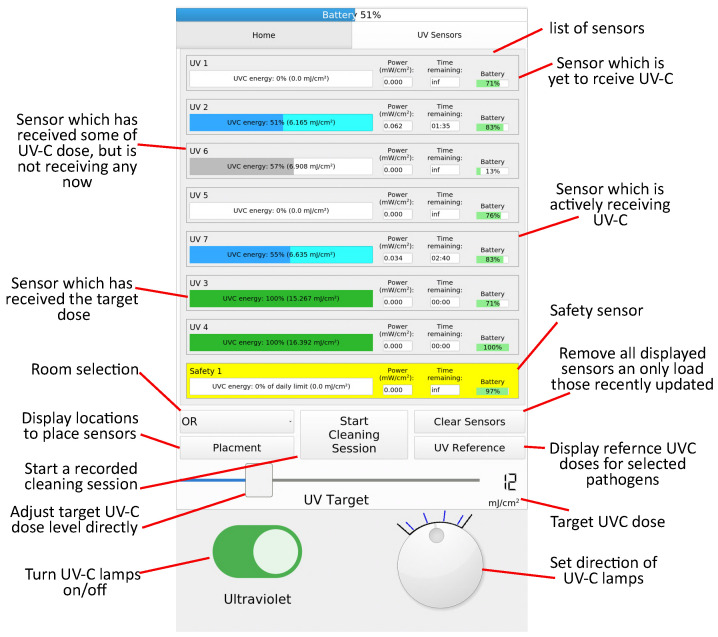
The robot user interface used to display the sensor information. The appearance of the progress bar adapts based on the state of the sensor allowing the user to more easily identify the sensor of interest, while 8 sensors are shown, additional sensors can be added arbitrarily without the need to modify the robotic system.

**Figure 6 sensors-23-02493-f006:**
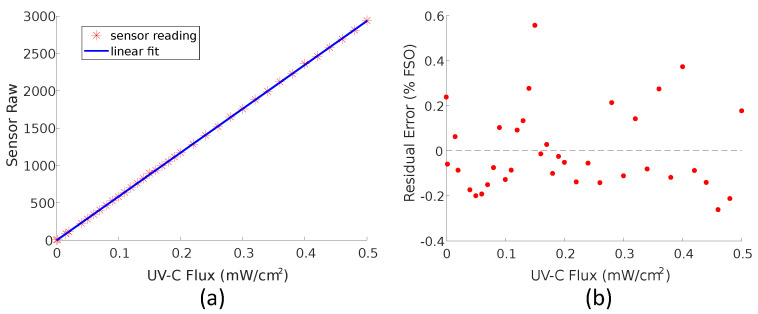
The results of the linearity tests. (**a**) shows the raw output of the ADC as a function of the UV-C radiant flux recorded with the reference meter. A linear model is fitted to the measured values. (**b**) shows the error between the linear model and the sensor output for each measurement. This is represented as a percentage of the full scale output (FSO) which for this purpose is taken as the reading at 0.5 mW/cm^2^.

**Figure 7 sensors-23-02493-f007:**
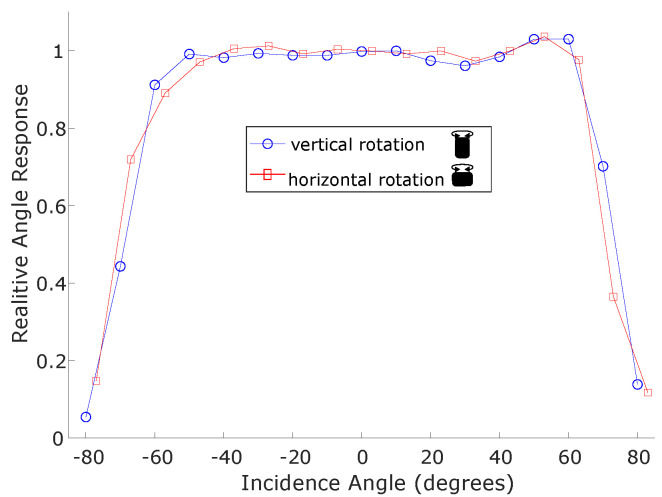
The results of cosine test, when the sensor is rotated in two orientations.

**Figure 8 sensors-23-02493-f008:**
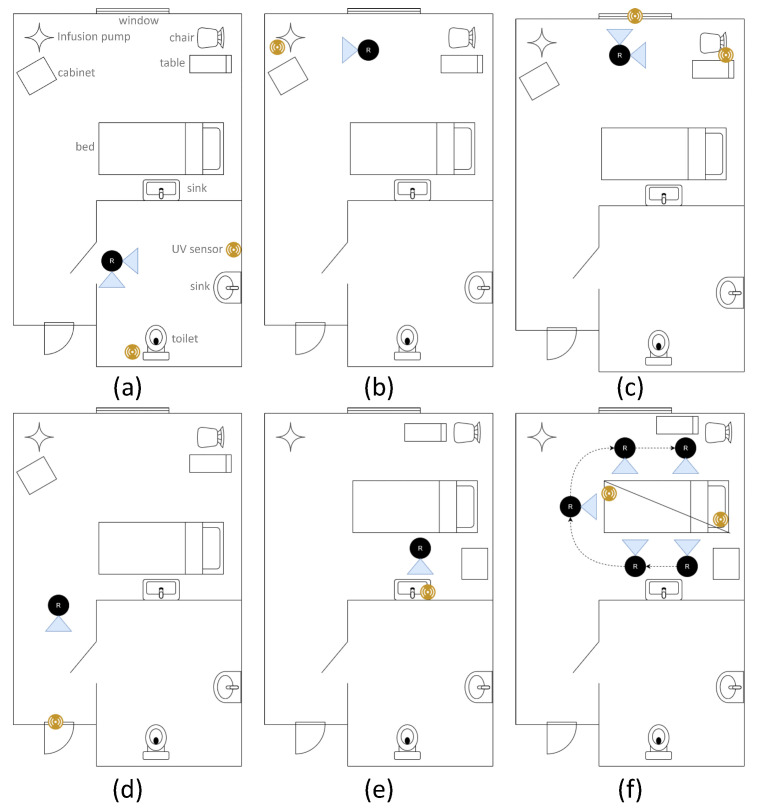
An illustration of an example operating procedure used during this trial. (**a**) shows the robot in its first position, in the bathroom. The blue triangles show the two directions in which the robot directs UV-C radiation. (**b**) shows a cabinet and infusion pump being treated. (**c**) shows the rooms window, chair and table being irradiated. (**d**) illustrates the entrance and bathroom door treatment. In (**e**) the cabinet and table are re-positioned and the sink is irradiated, while in (**f**) the bed is treated, including turning the mattress over.

**Figure 9 sensors-23-02493-f009:**
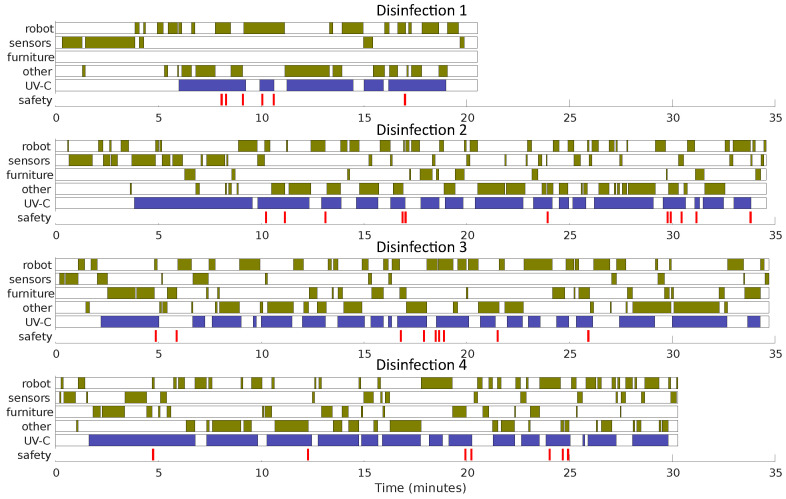
A time breakdown of four of the disinfection sessions with the robot and sensors which is approximated from the resulting video recording. Each session is broken into activities: “robot” records time directly interacting with the robot (controlling UV-C lamps or repositioning), “sensors” refers to time placing and moving sensors, “furniture” refers to time spent moving furniture in order for the robot to access parts of the room or to expose a different side of the furniture, “other” refers to time not occupied with the previous disinfection tasks which is available to do other activities such as manual cleaning or room preparation. In addition “UV-C” shows the times when the robot’s UV-C lamps were active and the “safety” markers refer to incidents when the safety sensor received UV-C irradiation.

**Table 1 sensors-23-02493-t001:** UV-C doses detected by the UV-C safety sensor worn by the robot operator and the number of exposures during each of the disinfection procedures. Note, that for UV-C irradiation at this wavelength, the occupational daily safety limit is 6 mJ/cm^2^ and the percentage of this value which the operator was exposed to is shown on the table. The final two procedures were not recorded. Furthermore, a breakdown of the duration of the procedures and the total time the UV-C lamps were on is provided.

	Safety Sensor Fluence (mJ/cm^2^)	Percentage of Daily Limit	Number of Exposures	Total Time with Lamps on (mm:ss)	Total Time for Procedure (mm:ss)
Disinfection 1	0.049	0.82%	6	10:56	20:31
Disinfection 2	0.122	2.03%	11	23:08	34:34
Disinfection 3	0.11	1.83%	9	21:12	34:43
Disinfection 4	0.161	2.68%	7	22:35	30:17
Disinfection 5	0.179	2.98%	-	-	27:26
Disinfection 6	0.314	5.23%	-	-	39:09

## Data Availability

Data sharing not applicable.

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
