# Peer review of "Development and Demonstration of a Wireless Ultraviolet Sensing Network for Dose Monitoring and Operator Safety in Room Disinfection Applications"

_sensors, 2023, doi:10.3390/s23052493_

Round 1

Reviewer 1 Report

It is interesting work by the author but I have some queries/suggestions which need to be addressed before the publication.

1) The title needs to be more clear.

2) Minamata convention is discouraging countries from the usage of Mercury lamps. Please comment on this in your paper.

3) I think it's better and much safer if we can control the robot remotely rather than a guy doing it along with a robot inside a room. Please comment.

4) I think decreasing the robot's size will help move the robot better inside the hospital room. Please comment critically on this.

5) Also, please put in the outlook that better design of hospital rooms for UVC robots will help in future.

Reviewer 2 Report

The article deals with an interesting topic. UVC disinfection has many advantages and has been shown to be effective in several studies, although UVC radiation is classified as carcinogenic by the IARC Agency. UVC radiation has not been included by the Environmental Protection Agency as a disinfectant on the "N" list. The article proposes a method to evaluate the usability and practicality of a mobile UVC disinfection system in the presence of personnel

Major Revision

Abstract

Line 5: the exposure to UVC radiation is regulated, not the UVC artificial radiation. Please correct the sentence

Line 9: Please, explain better materials and method in the abstract.

Key words

Please, define “BLE”.

Introduction

Line 28: Please, add reference.

Line 105: Please, add reference.

Materials and Methods

Lines 315-317: the sentence is not clear. Please rephrase.

Line 308: The terms “the sensor readongs” maybe are “the sensor readings”.

Line 356: The term “clinical testing”, it is not appropriate for the type of this study.

Line 361: Please, replace “Manual Operating Procedure” with “UVC- protocol.”

Line 364: The author can delete the part describing the cleaning protocol applied as terminal disinfection, because bactericidal efficacy is not evaluated in this study. Only the irradiation dose on exposed surfaces has been reported and the value recorded by the dosimeter is not influenced by the presence of bacterial load.

Line 366: The author reports that the robot was operated by a “researcher familiar”. Was there no personnel training during the experiment? Who is the end user of this device? It would have been appropriate to evaluate the usability of the device by the user personnel in a real scenario. Please explain this choice and mark it on the text.

Results

Figure 8: Please check the reference to f, it is reported again as e.

Line 405: The authors, sometimes use the term, “cleaning procedure” and in the Table 1 the term “disinfection “was used for UVC protocol. When the author refers to the UVC procedure, please change the term to disinfection.

Line 410: Please, explain how the percentage daily dose has been calculated and report it in the text.

Table 1: Please, correct the value of percentage of daily limit in the second line (disinfection2). Based on the data reported, the value should be 1.86%

Discussion

Line 434: Please, explain better the six different disinfection protocol, including “disinfection 1” and report the description in “Materials and Methods”, reporting how many sensors has been placed.

Conclusion

While these types of devices enhance the hygiene quality of the standard cleaning protocol, they are not considered completely safe for the operator.

Even if the authors report a radiation dose value lower than those foreseen by EU directive 2006/25/EC, they should consider how many disinfection procedures are performed by the the same operator, during the same working day. To ensure greater security, the use of multiple operators should be recommendable.

In this regard, it would also be advisable to evaluate the hourly cost of personnel based on the time saved between disinfection procedures.

According to the methods described by the author, time saving is not guaranteed as the operation of the device was interrupted several times due to the close distance reached by the operator during cleaning disinfection. For these reasons, the author should revise the concluding section, considering the above considerations. In our view, use in a high-turnover healthcare setting (e.g. radiology setting or operating theaters) may be considered as a possible application of this protocol.

Reviewer 3 Report

The manuscript describes the application of a network of ultraviolet sensors to monitor environmental and occupational exposure conditions.

The manuscript is generally acceptable. However, I offer the following comments:

The International Commission on Illumination uses the term UV-C and not UVC. Consideration should be given to using the internationally agreed designation.

The use of energy and power is incorrectly used interchangeably with fluence and fluence rate/flux. I suggest a complete review of the text to ensure that the correct terms are used.

Line 43. The decrease of UV-C flux with distance will depend on the geometry of the UV-C lamps. For the devices described in the manuscript, the flux is likely to decrease linearly with distance initially and then as one over the distance (inverse square law).

Line 44. I suggest avoiding the term "the light". The term "light" is incorrect for UV-C and the context implies that the sentence is referring to any optical radiation. It may be acceptable to replace "the light" with "optical radiation". Note that the use of the term "light" should be checked throughout the manuscript. In many cases, "optical radiation" would be more appropriate.

Line 105, the reference is missing (?).

Line 110, suggest adding "weighted" before "daily exposure limit".

Line 112, to avoid confusion, suggest adding "unweighted" before "daily limit".

Line 162, "high power" is not specific. Please add a value.

Line 181, should "method" be "device"?

Line 253, "brightness" is not an appropriate term for a UV-C source.

Line 308, "readongs" should be "readings"

Conclusions:

It should be pointed out that the devices worn by the operator are recording the fluence in the absence of clothing, gloves and face shield. Humans have evolved without natural exposure to UV-C and, as commented in the manuscript, there is insufficient information on the risks of repeated exposure, even if below the exposure limit. Therefore, suggesting repeated exposure in the room has an unknown risk.

It may be worth commenting on the impact of shielding sites with the sensor. What if there was a bio-active agent behind it? What about contamination of the sensor?

References:

I suggest checking the format of references 17, 21, 24 and 32

Round 2

Reviewer 1 Report

Accepted.

Author Response

Thank you again for your feedback

Reviewer 2 Report

Abstract, Line 15: it is not yet clear how the robot was used in the hospital setting (after a cleaning and disinfection protocol, with multiple exposure cycles, etc.)

Table 1: the value of percentage of daily limit in the second line (disinfection2) was not corrected

Line 391: manual disinfection procedure (manual positioning of the robot)

Line 496: add “but with the same radiation exposure characteristics”

Line 544: add “the use of this type of robot and its use by expert personnel”

Author Response

Thank you for your additional feedback. We have responded in red below

Abstract, Line 15: it is not yet clear how the robot was used in the hospital setting (after a cleaning and disinfection protocol, with multiple exposure cycles, etc.)

The following has been added/adapted

"Enhanced disinfection procedures can then be conducted as items in the room can be rearranged during the procedure to maximise the UV-C fluence delivered to otherwise inaccessible surfaces while allowing UV-C disinfection to occur in parallel with traditional cleaning. The system was tested for terminal disinfection of a hospital ward. During the procedure the robot was repeatedly manually positioned in the room by the operator, who used feedback from the sensors to ensure the desired UV-C dose was achieved, while also conducting other cleaning tasks. Analysis verified the practicality of this disinfection methodology, while highlighting factors which could affect its adoption."

The note on this being conducted after a clean isn't included here as this was a practical consideration for the trial. Ideally the robot's operation and cleaning would occur concurrently

Table 1: the value of percentage of daily limit in the second line (disinfection2) was not corrected

We should have been clearer on the error that was corrected here. The percentage of the daily limit was actually correct but the fluence value was mistranscribed (the incorrect value was 0.112 and it has been corrected to 0.122) 

Line 391: manual disinfection procedure (manual positioning of the robot)

Added this clarification to the text

"(where the robot is manually positioned by the operator)"

Line 496: add “but with the same radiation exposure characteristics”

added  "at a point with the same radiation exposure characteristics"

Line 544: add “the use of this type of robot and its use by expert personnel”

The line now reads

"As the risks associated with regular doses of UV-C below the daily limit is uncertain, the combination of personal protective equipment and active sensing offers redundancy for applications such as this which seek to use specialised hardware with trained personnel to allow UV-C disinfection alongside a person."